# Patterns and determinants of modern contraceptive discontinuation among women of reproductive age: Analysis of Kenya Demographic Health Surveys, 2003–2014

**Susan Ontiri**[1,2]*, **Vincent Were**[3], **Mark Kabue**[4], **Regien Biesma-Blanco**[2], **Jelle Stekelenburg**[2,5]

**1** Jhpiego, Johns Hopkins University Affiliate, Nairobi, Kenya, **2** Department of Health Sciences/Global Health, University of Groningen/University Medical Center Groningen, Groningen, The Netherlands, **3** Health Economics Research Unit, Kenya Medical Research Institute-Wellcome Trust, Nairobi, Kenya, **4** Jhpiego, Johns Hopkins University Affiliate, Baltimore, Maryland, United States of America, **5** Department of Obstetrics and Gynecology, Leeuwarden Medical Centre, Leeuwarden, The Netherlands

* Susan.Ontiri@jhpiego.org

**Data Availability Statement:** This study used the DHS Program's Kenya data sets for 2003, 2008-

## Abstract

### Objectives

This study aimed to examine patterns and determinants of modern contraceptive discontinuation among women in Kenya.

### Methods

Secondary analysis was conducted using national representative Kenya Demographic and Health Surveys of 2003, 2008/9, and 2014. These household cross-sectional surveys targeted women of reproductive age from 15 to 49 years who had experienced an episode of modern contraceptive use within five years preceding the surveys from 2003 (n = 2686), 2008/9 (n = 2992), and 2014 (5919). The contraceptive discontinuation rate was defined as the number of episodes discontinued divided by the total number of episodes. Weighted descriptive statistics, multivariable logistic regression analysis, and Cox proportional hazards analysis were used to examine the determinants of contraceptive discontinuation.

### Results

The 12-month contraceptive discontinuation rate for all methods declined from 37.5% in 2003 and 36.7% in 2008/9 to 30.5% in 2014. Consistently across the three surveys, intrauterine devices had the lowest 12-month discontinuation rate (6.4% in 2014) followed by implants (8.0%, in 2014). In 2014, higher rates were seen for pills (44.9%) and male condoms (42.9%). The determinants of contraceptive discontinuation among women of reproductive age in the 2003 survey included users of short-term contraception methods, specifically for those who used male condoms (hazard ratio [HR] = 3.30, 95% confidence interval [CI] = 2.13–5.11) and pills (HR = 2.68; 95CI = 1.79–4.00); and younger women aged 15–19 year (HR = 2.07; 95% CI = 1.49–2.87) and 20–24 years (HR = 1.94; 95% CI = 1.61–

09, 2014 (https://dhsprogram.com/data/available-datasets.cfm).

**Funding:** The authors received no specific funding for this work.

**Competing interests:** The authors have declared that no competing interests exist.

2.35). The trends in the most common reasons for discontinuation from 2003 to 2014 revealed an increase among those reporting side effects (p = 0.0002) and those wanting a more effective method (p<0.0001). A decrease was noted among those indicating method failure (p<0.0001) and husband disapproval (p<0.0001).

## Conclusions

Family planning programs should focus on improving service quality to strengthen the continuation of contraceptive use among those in need. Women should be informed about potential side effects and reassured on health concerns, including being provided options for method switching. The health system should avail a wider range of contraceptive methods and ensure a constant supply of commodities for women to choose from. Short-term contraceptive method users and younger women may need greater support for continued use.

## Introduction

The global focus on family planning program has been on expanding access to modern contraceptives, assuring method choice, overcoming barriers to use, and improving quality of care [1]. Recent global estimates reveal that 63% of women of reproductive age, 15 to 49 years, used some form of contraceptives in 2017, up from 54.8% in 1990 [2,3]. Family planning is a critical component of safe motherhood programs due to its direct and indirect effect on maternal mortality. The contribution of unintended pregnancies on maternal mortality and morbidity is well documented; studies have estimated that effective use of contraceptives could avert up to 44% of maternal deaths [4].

Most public health programs have traditionally targeted non-contraceptive users to understand their reasons for not using them and designed interventions that address the prevailing gaps. However, there has been less programmatic focus on assessing the level of satisfaction among current contraceptive users [5]. While these programs have resulted in an increase in contraceptive use in general, globally, 38% of contraceptive users discontinue use of a method within the first 12 months [6]. Whereas not all contraceptive discontinuation should be of concern, since the fertility desires of women change over time, discontinuation while still in need is a concern because it contributes substantially to the total fertility rate, unintended pregnancies, and induced abortions [7–10]. Furthermore, an analysis of Demographic Health Survey (DHS) data from 34 developing countries revealed that 38% of women estimated with the unmet need of family planning were prior method users who had discontinued use [6]. This underscores the importance of focusing on current users to ensure their fertility needs are met.

Some studies have demonstrated that certain socio-demographic characteristics—such as younger women, higher parity, and unmarried or not in a union—are the most likely determinants of discontinuation [8,9,11], and that discontinuation rates are higher among short-term method users compared to long-acting reversible contraceptive (LARC) users, such as intra-uterine devices (IUDs) and implants [8]. For instance, a multicountry DHS analysis revealed that the lowest 12-month discontinuation rate was for IUDs while the highest was for condoms. In the same study, pills, injectables, periodic abstinence, and withdrawal were discontinued by about 40% of users within the first 12 months of use [8].

Analysis of the DHS's calendar data derived from the women's questionnaire is the major source of information on contraceptive discontinuation; it contains robust historical data on episodes of contraceptive use, recalled by women month by month, five years preceding the survey [9]. While there are concerns about recall bias and the validity of calendar data based on its complexity and information, an analysis comparing various studies that used the calendar data and other forms of questionnaires established that calendar data performs just as well or better in terms of reliability and validity on capturing information on contraceptive use [12–14]. Moreover, due to lack of a robust longitudinal cohort data that is nationally representative, this calendar data, with its complexities, remains the best source of data for discontinuation analysis.

Several studies have documented that contraceptive discontinuation when in need of a method is a measure of the quality of family planning services because it can be addressed by improved counseling and instituting follow-up mechanisms [15–17]. Understanding factors that affect the discontinuation of modern contraceptive use is crucial to enable family planning programs to identify appropriate strategies to improve the continuous use of modern contraception [5].

A 2007 DHS multicountry analysis on contraceptive discontinuation included data from Kenya's 2003 DHS [8]. Since then, Kenya has invested in policies to improve the health care and the social environment that promotes increased use of modern contraceptives. In 2007, the country developed and launched its first-ever national reproductive health policy that sought to provide an enabling environment to increase equitable access and improve quality, efficiency, and effectiveness of service delivery at all levels [18]. In 2009, the country further expanded access to contraceptive methods through community-based distribution programs that allowed community health volunteers to provide family planning information and services, such as pills and injectables, to women in hard-to-reach areas [19]. This policy environment, among other interventions, contributed to an increase in the use of modern contraceptives, from 32% in 2003 to 53% in 2014 [20]. In the last two decades, Kenya's contraceptive method mix (percentage of current modern method users who use the particular method in question), has largely been driven by the uptake of short-term methods though there has been a gradual increase in the use of LARC methods particularly implants. Injectables' share of the method mix among contraceptive users has fluctuated from 45.6% in 2003, 55.0% in 2008/9, to 47.9% in 2014. Pills has been on a downward trend from 23.3% in 2003, 18.8% in 2008/9, to 14.1% in 2014. Notably, implants' share of the method mix use has risen from 5.3% in 2003, 4.8% in 2008/9, to 18.2% in 2014. IUD has changed from 7.5% in 2003, 4.0% in 2008/9, and 5.9% in 2014. Condom use has grown from 3.7% in 2003 to 7.9% in 2014 [20–22]. Despite the overall increase in modern contraceptive use over the last 15 years with changes in the method mix, discontinuation still occurs among one-third of contraceptive users in Kenya [20]. There has not been a subsequent analysis of 2008/9 and 2014 data to assess changes in the determinants and reasons for discontinuation with the increase in modern contraceptive prevalence rates and the corresponding shift in the method mix. Thus it is important to examine contraceptive dynamics such as discontinuation so that family planning programs can provide quality services that meets client needs [23]. Furthermore, such data would allow policymakers and program implementers to monitor progress toward achieving international development goals for family planning. This analysis examines the trends and determinants of contraceptive discontinuation using data from 2003, 2008/9, and 2014 Kenya Demographic Health Survey (KDHS). Kenya has not conducted a national demographic health survey since the KDHS 2014 that would provide additional information on changes in the contraceptive use dynamics.

## Materials and methods

Secondary data analysis was conducted based on the KDHS 2003, 2008/9, and 2014 datasets, which are publicly available through the DHS program website https://dhsprogram.com/data/available-datasets.cfm. The KDHS is a nationally representative, cross-sectional household surveys that used a two-stage multistage sampling approach, with a sampling frame based on the 1999 and 2009 national census in Kenya. These analyses used data from the women's questionnaires, which collect data from women of reproductive age (15–49 years) on a range of socio-demographic characteristics and reproductive history. The population included 8195 women (KDHS 2003), 8444 women (KDHS 2008/9), and 31079 women (KDHS 2014). These analyses were restricted to women who reported to have ever used a method of contraception in the five years preceding the survey and had complete contraceptive histories. The final sub-sample included in the analyses were 2686 (KDHS 2003), 2992 (KDHS 2008/9), and 5919 (KDHS 2014). Information on contraceptive use in the DHS is collected in the form of a reproductive calendar. It contains the past monthly history of reproductive events including births, pregnancies, terminations, and episodes of contraceptive use for the five years before the survey. Female respondents were asked about their contraceptive use for each month of the five years prior to the survey. For months in which a woman reported discontinuing the use of a method she was asked the main reason. Excluded from this study were women who had never used contraceptives, women who had indicated that they were pregnant at the time of the survey, infecund women who were self-reported, and women with incomplete contraceptive information.

### Variable definitions

**Exposure** is defined as the duration of use of a specific method within one episode of use. Exposure begins with an initial month of adoption. It would end with self-reported discontinuation or with the month of the interview if the contraceptive method was still being used at the time of the interview.

**Contraceptive discontinuation** is defined as starting contraceptive use and then stopping for any reason while still at risk of unintended pregnancy [24].

**Discontinuation while "in need,"** in this paper, this refers to women who are at risk of becoming pregnant, do not want to become pregnant, and are not using contraception [9]. This was operationalized from the reasons given for discontinuation as recorded in the contraceptive calendar data. Reasons for discontinuation including method failure, side effects, health concern, access, cost, wanted more effective method, inconvenient to use, and husband opposed, were considered to be discontinuation while in need, whereas wanting to become pregnant, infrequent sex/husband away, marital dissolution and menopausal were noted not to be in need.

### Statistical analysis

Our analysis focused on women who discontinued modern contraceptive methods (pills, IUDs, injectables, implants, and male condoms). Contraceptive methods that were discontinued on the month of the interview or two months prior were censored to avoid bias due to unrecognized pregnancy since many women do not realize they are pregnant in their first trimester [9]. The 2008/9 KDHS had missing data for reasons for contraceptive discontinuation, a variable used in the computation of the determinants, therefore, it was not included in the survival analysis.

To calculate the discontinuation rate and the number of episodes, the KDHS datasets for 2003 and 2014 were first converted into an event file to report episodes. The discontinuation

rate was calculated using a life table that generated the net discontinuation rates. The rate of discontinuation was calculated by dividing the number of episodes discontinued in a month by the total number of episodes that reached that duration. This was also calculated for 12 months' duration for each of the discontinuation reasons. An individual woman may contribute more than one episode to the calculation.

The contraceptive method was tabulated against the reasons for discontinuation and socio-demographic characteristics accounting for sampling weights. Frequencies and percentages were obtained for descriptive statistics. Twelve-month contraceptive discontinuation rates were calculated as shown in Table 2.

A Cox proportional hazard model was used to obtain hazard ratios (HR), 95% confidence interval (95%CI). In the survival analysis, method discontinuation was the dependent variable and the covariates included contraceptive method, age category, residence, education, marital status, religion, number of living children, and wealth quintile. Trend analysis comparing changes in discontinuation rates or proportions between survey years were compared using a Cochrane-Armitage trend test. Results with p-value <0.05 were considered statistically significant. In the trend analysis, the 2003 survey was used as a base and 2014 as the end line. Additional analyses were conducted to explore the profile of contraceptive users and determinants of contraceptive discontinuation among women in need. These results have been presented as supplementary tables for reference only (S1 and S2 Tables).

## Results

### Characteristics of the study population

The results shown in Table 1 present the profile of women who discontinued use of a modern contraceptive methods. The majority were aged 25–34 years, 54.9% in 2003, 48.8% in 2008, and 53.8% in 2014. There was an increase in discontinuation among women who were Protestant from 68.9% in 2003 to 75.3% in 2014. Most were married or living together with a partner (76.7% in 2003, 75.3% in 2008, and 77.8% in 2014); living in rural areas (72.3% in 2003, 73.2% in 2008, and 51.1% in 2014). A higher percentage had a primary education—60.6% in 2003, 57.9% in 2008, and 52.1% in 2014. Discontinuation in rural areas declined from 72.3% in 2003 to 51.1% in 2014 while it increased among urban residents from 27.7% in 2003 to 48.9% in 2014. There was an increase in the number of women who discontinued in the highest wealth quintile from 30.0% in 2003 to 31.5% in 2014.

### Contraceptive discontinuation rates at 12 months for specific methods in 2003 and 2014 in Kenya

Table 2 presents the 12-month discontinuation rates. Overall, the 12-month discontinuation rates for all methods decreased from 37.5% in 2003 to 36.7% in 2008 and 30.5% in 2014. The reduction between 2003 and 2014 surveys was statistically significant (trend p-value<0.0001). Condoms had the highest discontinuation rate in 2003 at 59.4%, followed by pills at 46.2%. Similar trends were observed in 2008/9 with highest rates for condoms the 60.2% and pills at 44.2%. In 2014, the survey results showed a slight variation with pills having the highest rate at 44.9%, followed by male condoms at 42.9%. Between 2003 and 2014, the method with the lowest discontinuation rate was IUDs at 12.4% in 2003 and 6.4% in 2014. Implants had the lowest rate in 2008/9 and IUDs at 6.4% in 2014. Between 2003 and 2014 there was a significant decline in discontinuation rates for male condoms (59.4% vs 42.9%, trend p-value<0.0001) and implants (15.3% vs 8.0%, trend p-value = 0.0204), discontinuation rates were not different for pills, injectables, and IUDs.

**Table 1. Profile of women who discontinued use of contraceptives at least three months before the survey.**

| | Categories | 2003 | | 2008/9 | | 2014 | |
|---|---|---|---|---|---|---|---|
| | | N | %* | n | %* | n | %* |
| Age categories (Years) | 15–19 | 157 | 9.8 | 109 | 5.7 | 150 | 4.6 |
| | 20–24 | 467 | 29.2 | 585 | 30.5 | 773 | 23.6 |
| | 25–34 | 879 | 54.9 | 939 | 48.8 | 1764 | 53.8 |
| | 35–49 | 99 | 6.2 | 289 | 15.0 | 595 | 18.1 |
| Marital status | Never Married | 225 | 14.1 | 285 | 14.8 | 388 | 11.8 |
| | Married | 1228 | 76.7 | 1447 | 75.3 | 2552 | 77.8 |
| | Single** | 149 | 9.3 | 191 | 9.9 | 342 | 10.4 |
| Residence | Urban | 444 | 27.7 | 515 | 26.7 | 1606 | 48.9 |
| | Rural | 1158 | 72.3 | 1407 | 73.2 | 1676 | 51.1 |
| Education level | None | 70 | 4.3 | 76 | 4.0 | 99 | 3.0 |
| | Primary | 971 | 60.6 | 1113 | 57.9 | 1711 | 52.1 |
| | Secondary | 432 | 27.0 | 546 | 28.4 | 1011 | 30.8 |
| | Tertiary | 129 | 8.1 | 188 | 9.8 | 461 | 14.0 |
| Religion | Catholic | 419 | 26.1 | 417 | 21.7 | 636 | 19.4 |
| | Protestant | 1100 | 68.9 | 1380 | 71.8 | 2472 | 75.3 |
| | Muslim | 60 | 3.7 | 92 | 4.8 | 112 | 3.4 |
| | No religion | 16 | 1.0 | 31 | 1.6 | 43 | 1.3 |
| | Other | 7 | 0.5 | 2 | 0.1 | 19 | 0.6 |
| Number of living children | None | 181 | 11.3 | 216 | 11.2 | 310 | 9.4 |
| | 1–2 | 727 | 45.4 | 858 | 44.6 | 1617 | 49.3 |
| | 3–4 | 438 | 27.3 | 565 | 29.4 | 889 | 27.1 |
| | 5+ | 246 | 16.1 | 284 | 14.8 | 466 | 14.2 |
| Fertility intention | Wants another child | 805 | 51.0 | 936 | 49.8 | 1686 | 52.1 |
| | Undecided | 55 | 3.5 | 50 | 2.6 | 72 | 2.2 |
| | No more | 718 | 45.5 | 891 | 47.4 | 1480 | 45.7 |
| Wealth Quintile | Poorest | 161 | 10.0 | 216 | 11.3 | 298 | 9.1 |
| | Poorer | 268 | 16.8 | 332 | 17.3 | 571 | 17.4 |
| | Middle | 316 | 19.7 | 376 | 19.5 | 649 | 19.8 |
| | Richer | 376 | 23.5 | 419 | 21.8 | 730 | 22.3 |
| | Richest | 481 | 30.0 | 579 | 30.1 | 1032 | 31.5 |

*Some figures may not add up to 100% because of rounding.

** Single refers to Divorced/Separated/Widowed.

**Table 2. Twelve-month contraceptive discontinuation rates for specific methods between 2003–2014 in Kenya.**

| Method | 2003 | 2008/9 | 2014 | Trend P-value |
|---|---|---|---|---|
| Pills | 46.2 | 44.2 | 44.9 | 0.5162 |
| Injectables | 31.8 | 29.6 | 30.9 | 0.5471 |
| IUD | 12.4 | 28.4 | 6.4 | 0.0502 |
| Implants | 15.3 | 10.2 | 8.0 | **0.0204** |
| Male Condoms | 59.4 | 60.2 | 42.9* | **<0.0001** |
| **All Methods** | **37.5** | **36.7** | **30.5** | **<0.0001** |

## Survival analysis of determinants of contraceptive discontinuation rates in Kenya, 2003 and 2014

The results of multivariable Cox proportion hazard regression model (Table 3) revealed that in the 2003 survey, the determinants of contraceptive discontinuation included methods of contraceptive used, age of the woman, and marital status. In the 2014 survey, the determinants of discontinuation rates were methods of contraceptive used and the age of the woman.

In the 2003 survey, compared to those who discontinued use of IUD, discontinuation was more likely among those who used pills (adjusted hazard ratio [aHR] = 2.67; 95% CI = 1.79–4.00), male condom (adjusted hazard ratio [aHR] = 3.30; 95% CI = 2.13–5.11) and injectables (aHR = 1.60; 95% CI = 1.07–2.37), but implant users were less likely to discontinue use (aHR = 0.42; 95% CI = 0.20–0.87). Similarly, in the 2014 survey, compared to those who discontinued IUD, users of pills (aHR = 5.25; 95% CI = 3.81–7.25), male condom (adjusted hazard ratio [aHR] = 3.80; 95% CI = 2.58–5.59) and injectables (aHR = 3.32; 95% CI = 2.40–4.58) had a higher hazard ratio of discontinuation.

In the 2003 survey, an increase in age was associated with a decreased hazard ratio of discontinuation. Women aged 15–19 years were more likely to discontinue use compared to those aged 35–49 years (aHR = 3.83, 95% CI = 2.85–5.14), and women aged 20–24 (aHR = 3.09, 95% CI = 2.47–3.87). The same pattern is observed in 2014 where an increase in age was associated decreasing hazard rates of discontinuation.

In the 2003 survey, separated/widowed women had a higher hazard ratio of discontinuation compared to women who were married (aHR = 1.20; 95% CI = 1.01–1.42). However, in the 2014 survey, there were no significant association between marital status and discontinuation (Table 3).

## Reasons for discontinuation in Kenya, 2003 to 2014

Table 4 shows the overall trend in the main reasons reported for discontinuation of all episodes among women in need of contraceptive between 2003 and 2014. In 2003, a total of 2297 women reported 3041 episodes while in 2014, 5029 women reported 6961 episodes.

From the analysis, side effects remain the primary reason for discontinuation in the two surveys, with a significant increase from 25.4% in 2003 (reported by 608 women) and 29.0% in 2014 (reported by 1385 women), observed between the two surveys (p = 0.0002). There was a significant reduction in women reporting becoming pregnant while using contraceptives due to method failure, 15.7% in 2003 and 10.8% in 2014; (p<0.0001) and among those reporting husband disapproval, 3.5% in 2003 and 1.4% in 2014; (p<0.0001). The analysis further revealed an increase in discontinuation for women who wanted a more effective method, 3.9% in 2003 and 8.7% in 2014; (p<0.0001).

## Trend analysis of method discontinuation rates by reasons in Kenya 2003 and 2014

Table 5 presents method discontinuation rates broken down by reasons for discontinuation. Among injectable users, there was a significant increase in the proportion of discontinuation due to a desire to switch to another method from 2003 to 2014 (7.1% in 2003 to 9.9% in 2014, trend p = 0.0027) and need for a more effective method (0.3% in 2003 to 1.9% in 2014, trend p = 0.0001). Even though pills and injectables contributed largely to discontinuation due to side effects, there was a significant reduction between the two surveys; discontinuation rates of pills due to side effects declined from 23.3% in 2003 to 15.7% in 2014 (p<0.0001) and injectables from 18.8% in 2003 to 14.3% in 2014 (p<0.0002). Similarly, a significant decrease in the

**Table 3. Survival analysis of determinants of contraceptive discontinuation rates in Kenya, 2003 and 2014.**

| | 2003 | | 2014 | |
|---|---|---|---|---|
| | Crude HR | Adjusted HR | Crude HR | Adjusted HR |
| **Contraceptive method** | | | | |
| Pills | 3.43(2.29–5.14)* | 2.67(1.79–4.00)* | 5.56(4.00–7.72)* | 5.25(3.81–7.25)* |
| Male Condom | 4.70(3.03–7.28)* | 3.30(2.13–5.11)* | 4.88(3.29–7.25)* | 3.80(2.58–5.59)* |
| Injectables | 2.09(1.41–3.11)* | 1.60(1.07–2.37)* | 3.71(2.69–5.12)* | 3.32(2.40–4.58)* |
| Implants | 0.48(0.23–0.99)* | 0.42(0.20–0.87)* | 1.23(0.82–1.83) | 1.15(0.77–1.71) |
| Intrauterine device | Ref | Ref | Ref | Ref |
| **Age category (years)** | | | | |
| 15–19 | 3.74(2.99–4.68)* | 3.83(2.85–5.14)* | 2.33(1.81–3.01)* | 2.07(1.49–2.89)* |
| 20–24 | 2.85(2.41–3.36)* | 3.09(2.47–3.87)* | 2.13(1.83–2.48)* | 1.93(1.58–2.36)* |
| 25–34 | 1.76(1.50–2.07)* | 1.92(1.58–2.33)* | 1.54(1.37–1.73)* | 1.44(1.25–1.66)* |
| 35–49 | Ref | Ref | Ref | Ref |
| **Residence** | | | | |
| Urban | 0.96(0.86–1.08) | 0.99(0.79–1.24) | 1.12(1.00–1.25)* | 1.08(0.94–1.25) |
| Rural | Ref | Ref | Ref | Ref |
| **Education** | | | | |
| No education | 1.36(1.04–1.79)* | 1.28(0.94–1.75) | 0.97(0.74–1.28) | 1.04(0.78–1.39) |
| Primary | 1.29(1.10–1.51)* | 1.04(0.85–1.27) | 0.86(0.70–1.07) | 0.91(0.73–1.13) |
| Secondary | 1.14(0.96–1.37) | 1.05(0.85–1.29) | 0.90(0.72–1.13) | 0.87(0.70–1.09) |
| Higher | Ref | Ref | Ref | Ref |
| **Marital status** | | | | |
| Never Married | 1.50(1.27–1.78)* | 0.93(0.76–1.13) | 1.31(1.05–1.64)* | 0.87(0.69–1.08) |
| Single** | 1.18(1.00–1.39)* | 1.20(1.01–1.42)* | 1.03(0.88–1.20) | 1.08(0.93–1.26) |
| Married | Ref | Ref | Ref | Ref |
| **Religion** | | | | |
| Catholic | 0.94(0.63–1.41) | 1.00(0.69–1.45) | 0.75(0.50–1.13) | 0.72(0.47–1.09) |
| Protestant | 0.92(0.62–1.38) | 1.03(0.72–1.48) | 0.82(0.55–1.21) | 0.79(0.52–1.18) |
| Muslim | 0.99(0.63–1.57) | 1.12(0.73–1.72) | 0.91(0.59–1.42) | 0.93(0.59–1.47) |
| No religion | Ref | Ref | Ref | Ref |
| Other | 0.88(0.47–1.63) | 1.00(0.49–2.00) | 1.55(0.84–2.86) | 1.28(0.69–2.37) |
| **Fertility intention** | | | | |
| Wants another child | 1.33(1.20–1.47)* | 0.95(0.84–1.06) | 1.32(1.20–1.46)* | 1.00(0.89–1.13) |
| Undecided | 1.25(0.95–1.63) | 1.01(0.77–1.31) | 0.77(0.58–1.04) | 0.75(0.55–1.02) |
| No more | Ref | Ref | Ref | Ref |
| **Number of living children** | | | | |
| 1–2 | 0.72(0.60–0.87)* | 1.03(0.82–1.30) | 0.66(0.50–0.89)* | 0.84(0.63–1.11) |
| 3–4 | 0.54(0.45–0.66)* | 1.01(0.78–1.31) | 0.52(0.39–0.69)* | 0.77(0.57–1.04) |
| 5+ | 0.48(0.39–0.58)* | 1.24(0.93–1.67) | 0.47(0.35–0.64)* | 0.85(0.61–1.18) |
| None | Ref | Ref | Ref | Ref |
| **Wealth quintile** | | | | |
| Poorest | 1.18(0.98–1.42) | 1.22(0.92–1.62) | 0.90(0.76–1.06) | 1.04(0.83–1.29) |
| Poorer | 1.14(0.97–1.34) | 1.16(0.89–1.52) | 0.91(0.78–1.06) | 1.01(0.83–1.23) |
| Middle | 1.04(0.90–1.20) | 1.07(0.84–1.37) | 0.95(0.82–1.11) | 1.07(0.88–1.31) |
| Richer | 0.96(0.83–1.11) | 0.99(0.79–1.25) | 0.84(0.72–0.98)* | 0.94(0.79–1.12) |
| Richest | Ref | Ref | Ref | Ref |

* means p-value<0.05

** Single refers to Divorced/Separated/Widowed.

**Table 4. Reasons for discontinuation while still in need.**

| | 2003 | | | 2014 | | | |
|---|---|---|---|---|---|---|---|
| Reasons | % of episodes* | No. of women | % of women | % of episodes* | No. of women | % of women | Episode trend test |
| Method failure | 15.7 | 336 | 14.6 | 10.8 | 501 | 10 | **<0.0001** |
| Husband disapproved | 3.5 | 79 | 3.5 | 1.4 | 56 | 1.1 | **<0.0001** |
| Side effects | 25.4 | 608 | 26.5 | 29.0 | 1385 | 27.5 | **0.0002** |
| Health concerns | 3.2 | 65 | 2.8 | na** | na | na | na |
| Access/availability | 1.8 | 45 | 1.9 | 0.8 | 44 | 0.9 | **<0.0001** |
| Wanted a more effective method | 3.9 | 63 | 2.7 | 8.7 | 442 | 8.8 | **<0.0001** |
| Inconvenient to use | 3.4 | 70 | 3 | 1.9 | 107 | 2.1 | **<0.0001** |
| Up to God/fatalistic | na | na | na | 0.1 | 6 | 0.2 | na |
| Cost too much | 1 | 23 | 1 | 0.8 | 46 | 0.9 | 0.3193 |
| **Total episodes** | **3041** | **2297** | | **6961** | **5029** | | |

*Percentages do not sum to 100% since women who discontinued while not in need of contraceptives, including those who reported they wanted to become pregnant/ experienced menopause (2003, 42.1% and in 2014, 46.5%) were excluded from the analysis.

na* implies the reason for discontinuation was not collected in the questionnaire for that particular year.

**Table 5. Difference in method discontinuation rate by reason among women in need of contraception in Kenya between 2003 and 2014.**

| | Method failure | | | Side effects/health concerns | | | Switch to another method | | | Wanted a more effective method | | | Other method related reasons* | | |
|---|---|---|---|---|---|---|---|---|---|---|---|---|---|---|---|
| Methods | 2003 | 2014 | P-value | 2003 | 2014 | P-value | 2003 | 2014 | P-value | 2003 | 2014 | P-value | 2003 | 2014 | P-value |
| Pills | 4.0 | 5.3 | 0.1318 | 23.3 | 15.7 | **<0.0001** | 12.4 | 19.8 | **<0.0001** | 2.4 | 6.8 | **<0.0001** | 4.7 | 2.2 | **0.0003** |
| Injectables | 1.0 | 1.7 | 0.0777 | 18.8 | 14.4 | **0.0002** | 7.1 | 9.9 | **0.0027** | 0.3 | 1.9 | **0.0001** | 2.6 | 1.2 | **0.0004** |
| IUD | 0.6 | 0.8 | 0.8373 | 7.9 | 4.2 | 0.1393 | 1.9 | 3.8 | 0.3469 | 0 | 0 | NA | 0 | 0.2 | 0.645 |
| Implants | 0.7 | 0.3 | 0.5351 | 12.0 | 6.6 | 0.0604 | 1.2 | 3.4 | 0.2723 | 0 | 0.1 | 0.7718 | 0 | 0.1 | 0.7718 |
| Condoms | 3.7 | 1.9 | 0.0536 | 0.3 | 0.8 | 0.3313 | 8.8 | 4.6 | **0.0031** | 3.8 | 2.3 | 0.1321 | 10.1 | 0.3 | **<0.001** |

*Includes: access/availability, inconvenient to use, and cost. P-values generated from Cochran trend test; IUD, intrauterine device.

level of discontinuation due to method switching was observed among condom users (8.8% in 2003 to 4.6% in 2014, p = 0.0031) users.

## Discussion

Our analysis has established that the contraceptive discontinuation rate in Kenya among modern method users significantly declined from 37.5% in 2003 to 30.5% in 2014. This rate is slightly lower than those observed in other sub-Saharan Africa countries, including Ghana (54.0%), Ethiopia (37.1%) Tanzania (37.7%), and Malawi (34.1%) [8,25,26].

The decline in contraceptive discontinuation rate observed between 2003 and 2014 may be attributed to the increase in the modern contraceptive prevalence rate—53% in 2014, up from 32% in 2003—with a corresponding increase in LARC uptake—13.3% in 2014 compared to 4.1% in 2003 [20,22]. Following the 2012 London Summit on Family Planning, the increased focus on family planning, globally and in Kenya, could explain the decrease in contraceptive discontinuation as attention was placed on scaling up of new methods, which expanded the method mix, and building the capacity of health care workers on the provision of contraceptive methods, particularly implants, and engaging the community to support family planning [27].

Results of the survival analysis established that the contraceptive methods used and the age of the woman were the main determinants of discontinuation. The 2003 survey also identified marital status (single) as an additional determinant. Discontinuation was more likely to be reported among users of short-term methods of contraceptives and among younger women. Additional exploration of the data to understand whether these determinants were different among women in need of contraceptive showed that the results were similar (supplementary file, S1 Table). These findings are consistent with studies done in Ethiopia and Senegal and in the DHS analysis of 60 countries in Africa, Asia, Eastern Europe, and Latin America that indicated short-term methods have the highest discontinuation rates [8,9,15,28]. In Kenya, use of short-term contraceptive methods—specifically injectables, pills, and condoms—among women of reproductive age, increased from 23.0% in 2003 to 36.4% in 2014 [20,22]. Removal of LARC methods such as IUDs and implants, requires a health care worker, which could explain the lower discontinuation rates, as compared to short-term contraceptive methods such as condoms, pills, and injectable contraceptives, which can be abandoned by the users without interaction with health care workers [9,29,30].

Our data presented as supplementary file, S2 Table indicated that LARC users were more likely to be older women and were more likely to be using the methods for limiting their family size, thus less likely to discontinue. Use of permanent contraceptive methods, particularly female sterilization was quite low at 3.2% in 2014 [20], hence more women who no longer desire to have children were more likely to use a LARC method, which confers longer-term protection as opposed to a short-term method that require frequent resupply.

The trend analysis revealed that side effects continue to be the leading reason for discontinuation among women, followed by method failure. These findings are corroborated in other studies conducted globally [7–9,11,26,31]. When coupled with results that indicate short-term method users are more likely to discontinue, we posit that discontinuation due to side effects occurred mostly among users of injectables and pills. This is corroborated by a study conducted in sub-Saharan Africa that reported that users of pills and injectables who experienced side effects that were not tolerable, mostly due to changes in bleeding patterns, discontinued use or switched to a method perceived to be more tolerable [32]. No modern contraceptive method is free of side effects, however, epidemiological studies conducted over the last four decades, largely in the United States and Europe, to evaluate the health effects associated with the use of these methods have established that the benefits of contraceptive use outweigh health risks [28]. Several studies have also indicated that the fear of side effects/health concerns whether perceived or real can influence a woman's decision to discontinue use of a contraceptive method, or deter potential new users [9,15,17,28]. These concerns need to be taken seriously and addressed.

Other notable reasons for contraceptive discontinuation were switching to another method or wanting a more effective method; this was especially true for women who used pills, injectables, and condoms. Women who use short-term method must make a conscious effort to maintain consistent use of their contraceptive and are therefore more prone to discontinuation [33]. Implants were not widely available in Kenya during the 2003 and 2008/9 surveys, which probably explains their low uptake of 1.7% and 1.9% respectively [21,22]. It is worth noting that between 2003 and 2014, implants contraceptive prevalence rate increased from 1.7% to 9.9%, which may explain the shift from other methods to implants [20]. Access to implants tremendously increased in Kenya, and other developing countries, following the 2012 London Summit, which resulted in commitments by countries, donors, and pharmaceutical companies to increase access to contraceptives, including implants, reduce the cost of commodities, and guarantee supplies of contraceptives [27]. Kenya implemented an implant scale-up program that saw the nationwide roll-out of the method as part of expanding the method mix, which

led to a significant shift, mostly observed among short-term method users, underscoring the desire by women to get a method that confers long-term protection. Additionally, studies have shown that, during the first year of typical use, LARC methods are more effective than short-term methods: implants are 120 times more effective than injectables, and 180 times more effective than pills; 6% of women using an injectable and 9% of women using pills experienced an unintended pregnancy during the first year of use compared to 0.05% of women with an implant and 0.8% of women with an IUD [34,35]. These realities could explain the reason why women want a more effective method. However, to get users' perspectives, additional studies need to be undertaken to understand women's desire to have more effective methods.

National family planning programs should intentionally put in place strategies to address contraceptive discontinuation. Side effects are a major concern among current and potential contraceptive users. As part of provision of quality family planning services, during the initiation of contraceptive use, women ought to be provided with information on potential side effects and options to consider, including method switching, when they experience intolerable side effects. Health education should be provided to women to allay potential fear of side effects.

Technological advancement is needed to support manufacturing of contraceptive technologies that are better tolerated by women. Expanding available contraceptive options ensures that women have a wide variety of methods to choose from if they are not satisfied with their current method.

The package of services offered by community health volunteers at the household-level should include follow-up with contraceptive users, especially short-term method users, to establish their level of satisfaction with their current method, to support users to ensure consistent use while in need, and to link women to health facilities when they experience concerns about the method to improve continuation. With the high mobile phone penetration of over 80% in Kenya, use of mhealth should also be embraced as a channel for provision of contraceptive information to women, including sending reminders to short-term method users to avoid an unintentional discontinuation [36].

One of the major strengths of our study was the use of large, nationally representative survey data, which makes the findings generalizable and applicable to countrywide policies and interventions. The multi-year analysis included data collected for 15 years; this allowed us to monitor changes in contraceptive use dynamics and the impact of policies and programs rolled out over the same period. It also provides an opportunity for future analysis. Nonetheless, there are certain limitations to our study. Although the findings were interesting and insightful, DHS data are cross-sectional, hence causal relationships cannot be established. DHS also do not collect data on the quality of family planning services that would allow us to assess the relationship between discontinuation and quality of care received. Our analysis used the contraceptive calendar data that is collected for five-year period prior to the survey. Women were asked to recall their contraceptive use history, month-by-month basis, which could result to recall bias. Our findings should be interpreted in the context of these limitations. However, the significance of findings can be improved with a future analysis when less biased data becomes available. Another limitation is that data collected only one main reason for discontinuation, while women might have multiple reasons for discontinuation.

## Conclusion

Our study reports wide variation in contraceptive discontinuation rates by method, with lower probabilities of discontinuation of highly effective methods, such as the long-acting reversible methods. The study also demonstrates a significant reduction in contraceptive discontinuation

between 2003 and 2014. With the increase in the use of modern contraceptives, programs should monitor trends in contraceptive use dynamics, including discontinuation and method switching. Public health programs should strengthen service quality, including the supply chain system, and enhance the provision of information that is client-centered as part of a rights-based approach for family planning services. Despite the improvement in service quality, some women may still choose to discontinue use of a contraceptive while in need of a method, which underscores the need to expand available options.

Longitudinal studies to assess women's contraceptive dynamics, including discontinuation and method switching, can better inform programs and address contraceptive discontinuation. More studies should also be conducted to understand the level of information on side effects that women are provided by health care providers during initiation of contraceptive use and the impact the information might have on contraceptive continuation. In addition, further studies are needed to understand whether the side effects that women report in the DHS, reflects their lived experience of FP use or their fears and perception since DHS does not distinguish between the two. Understanding and responding to why women discontinue use of a method while still in need should increase use of modern contraceptives, hence reduce unintended pregnancies and, in turn, maternal morbidity and mortality, improving the lives of women and their families.

## Supporting information

**S1 Table. Survival analysis of determinants of contraceptive discontinuation while in need in Kenya 2003 and 2014.**
(PDF)

**S2 Table. Profile of ever users of family planning methods.**
(PDF)

## Acknowledgments

The authors acknowledge the DHS Program and ICF for providing us with access to the datasets on our request. We are grateful to Elizabeth Thompson, Jhpiego Baltimore who edited the manuscript.

## Author Contributions

**Conceptualization:** Susan Ontiri, Mark Kabue, Regien Biesma-Blanco, Jelle Stekelenburg.

**Data curation:** Susan Ontiri, Vincent Were.

**Formal analysis:** Susan Ontiri, Vincent Were.

**Investigation:** Susan Ontiri.

**Methodology:** Susan Ontiri.

**Supervision:** Regien Biesma-Blanco, Jelle Stekelenburg.

**Writing – original draft:** Susan Ontiri.

**Writing – review & editing:** Susan Ontiri, Vincent Were, Mark Kabue, Regien Biesma-Blanco, Jelle Stekelenburg.

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
