## [Decision Letter · Decision Letter 0]

14 Aug 2020

PONE-D-20-19927

Patterns and determinants of modern contraceptive discontinuation among women of reproductive age: Analysis of Kenya Demographic Health Surveys, 2003–2014

PLOS ONE

Dear Dr. Ontiri,

Thank you for submitting your manuscript to PLOS ONE. After careful consideration, we feel that it has merit but does not fully meet PLOS ONE’s publication criteria as it currently stands. Therefore, we invite you to submit a revised version of the manuscript that addresses the points raised during the review process.

We look forward to receiving your revised manuscript.

Kind regards,

Catherine S. Todd

Academic Editor

PLOS ONE

Journal Requirements:

Additional Editor Comments (if provided):

In addition to the points raised by the reviewers, I recommend making both the Introduction and Discussion more concise and reading through to edit areas with awkward phrasing (such as over-use of the word "majority" in the first part of the Results or using colloquial expressions like "backed-up"). Please clarify how "in need" was defined for the purposes of this analysis when determining which women were retained. While the focus of the paper is to report discontinuation, it is difficult to put these rates into perspective without also having data for method prevalence/use so please add this to the Results. Last, I would like to see greater nuance in contextualizing discontinuation rates. Were women who used the IUCD older overall and thus using the IUCD to limit rather than space? These women may have different thresholds or reasons for discontinuation as compared to women who are using COCs provided at the community level. For the expanded community-level provision of some short-acting methods, how likely is supply chain to factor into continuation vs. actual rejection of the method? Overall, this is a well-written manuscript and if the issues raised here and with the reviewers are addressed, should soon be acceptable for publication.

Reviewers' comments:

Reviewer's Responses to Questions

**Comments to the Author**

1. Is the manuscript technically sound, and do the data support the conclusions?

Reviewer #1: No

Reviewer #2: Yes

2. Has the statistical analysis been performed appropriately and rigorously? 

Reviewer #1: I Don't Know

Reviewer #2: Yes

3. Have the authors made all data underlying the findings in their manuscript fully available?

Reviewer #1: Yes

Reviewer #2: Yes

4. Is the manuscript presented in an intelligible fashion and written in standard English?

Reviewer #1: Yes

Reviewer #2: Yes

5. Review Comments to the Author

Reviewer #1: This is a nice paper that utilized the complex DHS contraceptive calendar data to look at trends in contraceptive discontinuation rates in Kenya over three surveys. While I think the overall analysis and interpretation are sound, there are several key areas for the authors to improve, outlined below. The main issues are

1) the DHS dataset is complex, while also having potential concerns about validity given the reliance on 5 year recall, month-by-month for women. The authors need to explain this in a little more detail to both help the reader understand the complexity and to address head-on concerns about the validity of the interpretation.

2) the authors need to describe the statistical test and results they did to compare the trend of contraceptive discontinuation rates across the surveys.

Line 95: Change the sentence “Kenya’s 2003 DHS of 2003” to not repeat the year.

Overall, I think the introduction did a good job of providing important context to this study, however it becomes repetitive in the third paragraph (lines 65-72 state the same concept—contraceptive discontinuation while not wanting pregnancy—in multiple ways), and does not describe the strengths and weaknesses of the contraceptive calendar to allay readers’ potential concerns about the strength of the data.

Materials and Methods: Please add a sentence about how contraceptive calendar data is collected/organized (recall of family planning need and contraceptive use for each month for the previous 5 years).

Line 145/6: The sentence. “the datasets for 2003 and 2014 were first converted into an event file because the data was calendar data” is not clear.

Results:

Line 176-177: Could the authors calculate the statistical significance of the decrease in contraceptive discontinuation across the three time periods? This is also stated as the third sentence in the discussion (“no significant change”, but no discussion of whether a statistical test was used).

Table 2 would be improved with actual numbers, so we can see what proportion of the population was using each method.

Table 3: The ** and *** are not defined on the table.

Table 4: The authors need to describe the decision to classify “it’s up to god, or fatalism” as an effect modifier and the rationale for excluding it from the analysis.

Table 5 would be more accurately titled something like “Difference in method method continuation rates by reason, among women in need of contraception in Kenya between 2003 and 2014.”

Discussion:

Line 240: “A declined in contraceptive discontinuation between 2008 and 2014 was observed” seems to belie the preceding statement, lines 237-238, stating that no significant change was observed between 2003 and 2014—there are no statistical tests shown that support those statements.

Reviewer #2: Comments on PONE-D-20-19927

This paper addresses an important topic. It is well-written, nicely framed, and uses appropriate analytical methods. It was a pleasure to read. I have two primary comments that I raise for the authors’ consideration in their treatment of the discussion/conclusion and abstract, followed by several very minor comments.

Recommendations to prioritize short-term method users and young women seem to come from the finding that these users discontinue at higher rates than users of other methods/older women. However, discontinuation is not necessarily bad or a signal that something in wrong. Just discontinue while still in need (as the authors themselves point out in the introduction). There may be a selection effect: young women choosing condoms because it is easy to discontinue these methods when needs change. Focusing on these women/long-term methods writ large could actually risk ignoring their needs and impairing informed choice. This is more nuanced than is expressed in the abstract (the discussion does a slightly better job of this). I like the focus on resupply of short-term methods.

Similarly, the recommendation to focus on counseling because of the increasing share of reasons for discontinuation being side effects. 1. Did the authors analyze whether the relative increase in side effects as a reason is accompanied or not by an absolute increase in this reason? I.e. are more women discontinuing due to side effects? (Same Q for wanting a more effective method). I’m wary of responding to concerns of side effects with better counseling about side effects. There is a large literature (more developed in high-income countries, but existing globally) about the medical establishment discounting women’s reports of pain and other symptoms. There is also a literature about women’s experiences with side effects. The DHS measure “side effects/health concerns” cannot distinguish whether it is women’s experiences or fears of side effects. We should be careful not to assume that it’s all fears in women’s heads. How do the recommendations to enhance counseling, particularly on side effects, respond to women’s needs if it is women’s experiences of side effects prompting discontinuation? What other recommendations might we consider to ensure that women who experience side effects with their selected method nonetheless have their contraceptive needs met? The issues of counselling and side effects should be a little more developed in the discussion and conclusions section.

Minor comments:

Data are available through The DHS Program website. MEASURE DHS was an old contract and references to that name should be updated. Likewise, ICF International is now ICF.

Proofread: In at least one instance, the 2008-09 Kenya DHS appears as “2008/8” (p2, line 27).

Do you want more consistency between the terms “family planning” and “contraception” (or modern contraception) or is this distinction intentional?

Is uptake distinguishable from use? I believe, when discussing prevalence, use may be the more accurate term (p5, line 89).

P6, line 122: Do you mean “adoption” instead of “application”?

P7, line 126: It would more consistent to use “episodes” in lieu of “incidents”.

Can Table 1 be formatted to have some separation between or better alignment between the N and (%)? It is a little difficult to read as is.

When discussing changes in 12-month discontinuation rates across surveys (p10), it may be worth noting any substantial shifts in the method mix over that time period.

Please use aHR consistently. On p11, lines 199-200 refer to the aHR, but have it labeled HR, making it unclear whether it is the crude or adjusted hazard rate, without referring to the table. (Alternately, the crude HRs could be removed from Table 3.)

In Table 4 (p13), the na* on the row for “up to God/fatalistic” is in bold instead of plain text.

Presentation of data on reasons for discontinuation (p13) exclude all fertility/need-based reasons. I assume that these reasons make up the difference between the sum of the reasons presented in Table 4 and 100%, i.e. that the authors are presenting the relative share of each reason to all reasons for discontinuation. This should be clarified, particularly if I am wrong in this assumption.

In the discussion (p16), the authors summarize their results in terms of likelihoods. While there is a correspondence between likelihoods and hazards or hazard ratios, technically, the authors calculated the latter, not the former. I’m not sure this is a point worth getting worked up about, though.

6. PLOS authors have the option to publish the peer review history of their article (what does this mean?). If published, this will include your full peer review and any attached files.

Reviewer #1: No

Reviewer #2: **Yes: **Dr. Kerry L.D. MacQuarrie

---

## [Author Response · Author response to Decision Letter 0]

25 Sep 2020

Re: Revision of Manuscript “Patterns and determinants of modern contraceptive discontinuation among women of reproductive age: Analysis of Kenya Demographic Health Surveys, 2003–2014”

This paper addresses an important topic. It is well-written, nicely framed, and uses appropriate analytical methods. It was a pleasure to read. I have two primary comments that I raise for the authors’ consideration in their treatment of the discussion/conclusion and abstract, followed by several very minor comments.

1. Recommendations to prioritize short-term method users and young women seem to come from the finding that these users discontinue at higher rates than users of other methods/older women. However, discontinuation is not necessarily bad or a signal that something in wrong. Just discontinue while still in need (as the authors themselves point out in the introduction). There may be a selection effect: young women choosing condoms because it is easy to discontinue these methods when needs change. Focusing on these women/long-term methods writs large could actually risk ignoring their needs and impairing informed choice. This is more nuanced than is expressed in the abstract (the discussion does a slightly better job of this). I like the focus on resupply of short-term methods.

Response: We take cognizant of the issue raised. We conducted further exploratory analysis and established that the determinants of discontinuation among women who need contraceptive are still the same (short-term method users and younger women). These results are provided as supplementary files and not included in the main paper. Use of short-term methods was associated with higher rates of discontinuation. 

We have revised our recommendation as indicated in this excerpt from the abstracts as follows- “We recommend that Family planning programs should focus on improving service quality to strengthen the continuation of contraceptive use among those in need. Women should be informed about potential side effects and reassured on health concerns, including being provided options for method switching. The health system should avail a wider range of contraceptive methods and ensure a constant supply of commodities for women to choose from. Short-term contraceptive method users and younger women may need greater support for continued use.”

2. Similarly, the recommendation to focus on counseling because of the increasing share of reasons for discontinuation being side effects. 1. Did the authors analyze whether the relative increase in side effects as a reason is accompanied or not by an absolute increase in this reason? I.e. are more women discontinuing due to side effects? (Same Q for wanting a more effective method).

Response: Based on this comment, we have added a column in Table 4 to indicate the number of women contributing to these episodes and proportion of women. The increased share of reasons for discontinuation is accompanied by increased number of episodes (and proportion of women reporting those episodes. 

3. I’m wary of responding to concerns of side effects with better counseling about side effects. There is a large literature (more developed in high-income countries, but existing globally) about the medical establishment discounting women’s reports of pain and other symptoms. There is also a literature about women’s experiences with side effects. The DHS measure “side effects/health concerns” cannot distinguish whether it is women’s experiences or fears of side effects. We should be careful not to assume that it’s all fears in women’s heads. How do the recommendations to enhance counseling, particularly on side effects, respond to women’s needs if it is women’s experiences of side effects prompting discontinuation? What other recommendations might we consider to ensure that women who experience side effects with their selected method nonetheless have their contraceptive needs met? The issues of counselling and side effects should be a little more developed in the discussion and conclusions section.

Response: This is well noted. We have now qualified the recommendation on counseling to include counseling on other available methods to encourage women to switch when and if they experience side effects or health concerns. In the discussion section, we have alluded to the fact that women are not given adequate information on the potential side effects and what to do when they experience them and provided a case for method switching where possible. Furthermore, we have also added a recommendation on investing more in contraceptive technology to provide and array of options that are better tolerated by women. We also do appreciate that some of the reported side effects may be based on fears. On this, we have now recommended information to be provided to women to allay their fears which include dispelling myths and misconceptions to ensure usage while in need. And furthermore, we have recommended additional studies to explore whether the side effects that women cite is based on their experience or their fears. 

Minor comments:

4. Data are available through The DHS Program website. MEASURE DHS was an old contract and references to that name should be updated. Likewise, ICF International is now ICF.

Response: This is noted and have been revised accordingly as follows: The datasets used in the analysis are publicly available and can be accessed online on The DHS Program website https://dhsprogram.com/datasets

5. Proofread: In at least one instance, the 2008-09 Kenya DHS appears as “2008/8” (p2, line 27).

Response: This has been changed in line 28

6. Do you want more consistency between the terms “family planning” and “contraception” (or modern contraception) or is this distinction intentional?

Response: We have revised the write up to ensure consistency between the terms family planning and modern contraceptives. We have now dropped contraception, and adopted use of family planning when referring to the services that allow individuals to achieve desired birth spacing and family size, and timing of pregnancy, and contraceptive (or modern contraceptives) when referring to the actual methods such as IUD, implants that individuals use.

7. Is uptake distinguishable from use? I believe, when discussing prevalence, use may be the more accurate term (p5, line 89).

Response: That has been revised, and we have adopted the term use (line 100)

8. P6, line 122: Do you mean “adoption” instead of “application”?

Response: This has been changed to adoption (line 146)

9. P7, line 126: It would more consistent to use “episodes” in lieu of “incidents”.-

Response: This section where this comment was made (definition of contraceptive discontinuation rate) has been revised after we noticed repetition of this definition in statistical analysis section. In the statistical analysis section, lines 161-167 we have used “episodes” as opposed to “incidents” for uniformity. 

10. Can Table 1 be formatted to have some separation between or better alignment between the N and (%)? It is a little difficult to read as is. 

Response: This has been revised.

11. When discussing changes in 12-month discontinuation rates across surveys (p10), it may be worth noting any substantial shifts in the method mix over that time period.

Response: This analysis was done by DHS, and we did not repeat it in our results since our paper is focusing on discontinuation. However, we have included a paragraph in our introduction section on to indicate the shifts in the method mix (lines 100-112. This has further been referenced in our discussion (lines 286-287, 295, 317-319

12. Please use aHR consistently. On p11, lines 199-200 refer to the aHR, but have it labeled HR, making it unclear whether it is the crude or adjusted hazard rate, without referring to the table. (Alternately, the crude HRs could be removed from Table 3.)-

Response: This has been revised accordingly in lines 222-223

13. In Table 4 (p13), the na* on the row for “up to God/fatalistic” is in bold instead of plain text.

Response: This has been revised

14. Presentation of data on reasons for discontinuation (p13) exclude all fertility/need-based reasons. I assume that these reasons make up the difference between the sum of the reasons presented in Table 4 and 100%, i.e. that the authors are presenting the relative share of each reason to all reasons for discontinuation. This should be clarified, particularly if I am wrong in this assumption.- 

Response: The percentages don’t add up to 100% since women who were not in need of contraceptives (2003, 42.1% and in 2014, 46.5%) were excluded from the analysis. These included women who reported they wanted to become pregnant/menopause. 

15. In the discussion (p16), the authors summarize their results in terms of likelihoods. While there is a correspondence between likelihoods and hazards or hazard ratios, technically, the authors calculated the latter, not the former. I’m not sure this is a point worth getting worked up about, though. 

Response: The language has been revised to report hazard ratios for consistency. 

16. Additional Editor Comments (if provided):

In addition to the points raised by the reviewers, I recommend making both the Introduction and Discussion more concise and reading through to edit areas with awkward phrasing (such as over-use of the word "majority" in the first part of the Results or using colloquial expressions like "backed-up"). 

Response: We have revised the introduction and discussion section to be more concise and removed the awkward phrases. 

17. Please clarify how "in need" was defined for the purposes of this analysis when determining which women were retained. 

Response: This has been defined in the variable section as women who are at risk of becoming pregnant, do not want to become pregnant, and are not using contraception. A reference to this definition has been provided (lines 151-152)

18. While the focus of the paper is to report discontinuation, it is difficult to put these rates into perspective without also having data for method prevalence/use so please add this to the Results. 

Response: As mentioned in our response to comment 11, this analysis was done by DHS, and we didn’t not repeat it in our results since our paper was focusing on discontinuation. However, we have included a paragraph in our introduction on the contraceptive prevalence use and method mix, which has also been referenced in our discussion to assist during the interpretation of the discontinuation rate

19. Last, I would like to see greater nuance in contextualizing discontinuation rates. Were women who used the IUCD older overall and thus using the IUCD to limit rather than space? These women may have different thresholds or reasons for discontinuation as compared to women who are using COCs provided at the community level. For the expanded community-level provision of some short-acting methods, how likely is supply chain to factor into continuation vs. actual rejection of the method? Overall, this is a well-written manuscript and if the issues raised here and with the reviewers are addressed, should soon be acceptable for publication.

Response: We have further conducted additional analysis to show the socio-demographic profile of women who used various methods in 2003 and 2014 DHS, which has been presented as supplementary file 1. This analysis has established that most of the LARC users (implants and IUDs) were likely to be older women, hence most likely using the method for limiting. We have further included this aspect in our discussion section, lines 292-297.

The comment on the expanded community-based distribution is valid. It is true that supply chain issues especially on availability of the commodities has an impact on continuation or rejection of a method; if women feel they might be using a method that doesn’t have consistent supply they might opt out. However, this reason did not feature in our analysis. In our recommendation, we have made a case to strengthen service quality which includes the supply chain system both at facility and community level.

20. Reviewer #1: This is a nice paper that utilized the complex DHS contraceptive calendar data to look at trends in contraceptive discontinuation rates in Kenya over three surveys. While I think the overall analysis and interpretation are sound, there are several key areas for the authors to improve, outlined below. The main issues are

1) the DHS dataset is complex, while also having potential concerns about validity given the reliance on 5 year recall, month-by-month for women. The authors need to explain this in a little more detail to both help the reader understand the complexity and to address head-on concerns about the validity of the interpretation.

Response: Additional information on the potential bias and validity concern on the use of DHS calendar data has been provided in the introduction section lines 75-84; and the limitation in lines 363-369.

21. 2) the authors need to describe the statistical test and results they did to compare the trend of contraceptive discontinuation rates across the surveys.

Response: In line 177 we have stated that we used Cochrane-Armitage trend test to compare discontinuation rates across the survey years.

22. Line 95: Change the sentence “Kenya’s 2003 DHS of 2003” to not repeat the year. 

Response: This change has been made in line 92

23. Overall, I think the introduction did a good job of providing important context to this study, however it becomes repetitive in the third paragraph (lines 65-72 state the same concept—contraceptive discontinuation while not wanting pregnancy—in multiple ways), and does not describe the strengths and weaknesses of the contraceptive calendar to allay readers’ potential concerns about the strength of the data.

Response: The introduction section has been revised to remove the repetition. Additional details on the strengths and weaknesses of the contraceptive calendar has been provided in the introduction section, lines 75-84; an analysis comparing various studies that have used the calendar data and other forms of questionnaires, have established that the calendar data performs just as well or better in terms of reliability and validity on capturing information on contraceptive use

24. Materials and Methods: Please add a sentence about how contraceptive calendar data is collected/organized (recall of family planning need and contraceptive use for each month for the previous 5 years). 

Response: This has been added from lines 77-78 and 364-366

25. Line 145/6: The sentence. “the datasets for 2003 and 2014 were first converted into an event file because the data was calendar data” is not clear. 

Response: This has been revised to indicate that the calendar data was converted into an event to enable calculation of discontinuation rates (lines 161-163)

26. Results:

Line 176-177: Could the authors calculate the statistical significance of the decrease in contraceptive discontinuation across the three time periods? This is also stated as the third sentence in the discussion (“no significant change”, but no discussion of whether a statistical test was used).

Response: Table 2 has been revised to include statistical tests comparing the average trend from 2003, 2008 and 2014. The trend test assumes linearity of change from 2003 to 2014, hence reported as net change. We have further corrected the term ‘no significant change’ to ‘significant decrease’ after the statistical test indicate p-value <0.0001

27. Table 2 would be improved with actual numbers, so we can see what proportion of the population was using each method. We have included the actual number of episodes for each method. 

Response: We agree that Table 2 would have been improved if we reported the number of episodes or women. However, we note the complexity of reporting this. First, the calculation of 12-month discontinuation for each woman differs because each woman had different start and end dates for a 12-month duration for each episode. Secondly, some women discontinue multiple methods within the 12-month of the start of each episode and accounting for multiple discontinuations from multiple switchers, become more complex. We, therefore, keep the discontinuation rates without the proportion of the population of women

28. Table 3: The ** and *** are not defined on the table. 

Response: This has been revised and defined. The asterisk * to refer to significant results with p-values <0.05

29. Table 4: The authors need to describe the decision to classify “it’s up to god, or fatalism” as an effect modifier and the rationale for excluding it from the analysis.

Response: There was an error in our definition of the asterisk as an effect modifier which has since been revised. The * implied that the data on reason (It’s up to God or fatalism) was not collected in the 2003 survey. 

30. Table 5 would be more accurately titled something like “Difference in method continuation rates by reason, among women in need of contraception in Kenya between 2003 and 2014.” 

Response: The title of this table has been revised to reflect the proposed change.

31. Discussion:

Line 240: “A declined in contraceptive discontinuation between 2008 and 2014 was observed” seems to belie the preceding statement, lines 237-238, stating that no significant change was observed between 2003 and 2014—there are no statistical tests shown that support those statements.

Response: This has been revised based on the additional trend analysis that was conducted that indicates a significant change was observed (lines 264-265)

---

## [Decision Letter · Decision Letter 1]

12 Oct 2020

PONE-D-20-19927R1

Patterns and determinants of modern contraceptive discontinuation among women of reproductive age: Analysis of Kenya Demographic Health Surveys, 2003–2014

PLOS ONE

Dear Dr. Ontiri,

Thank you for submitting your manuscript to PLOS ONE. After careful consideration, we feel that it has merit but does not fully meet PLOS ONE’s publication criteria as it currently stands. Therefore, we invite you to submit a revised version of the manuscript that addresses the points raised during the review process.

We look forward to receiving your revised manuscript.

Kind regards,

Catherine S. Todd

Academic Editor

PLOS ONE

Additional Editor Comments (if provided):

My thanks to the authors for responding to most of the reveiwers' comments. Please kindly address the minor edits requested by Reviewer 1 and this manuscript will then be acceptable for publication.

Reviewers' comments:

Reviewer's Responses to Questions

**Comments to the Author**

1. If the authors have adequately addressed your comments raised in a previous round of review and you feel that this manuscript is now acceptable for publication, you may indicate that here to bypass the “Comments to the Author” section, enter your conflict of interest statement in the “Confidential to Editor” section, and submit your "Accept" recommendation.

Reviewer #1: All comments have been addressed

Reviewer #2: (No Response)

2. Is the manuscript technically sound, and do the data support the conclusions?

Reviewer #1: (No Response)

Reviewer #2: Yes

3. Has the statistical analysis been performed appropriately and rigorously? 

Reviewer #1: Yes

Reviewer #2: Yes

4. Have the authors made all data underlying the findings in their manuscript fully available?

Reviewer #1: Yes

Reviewer #2: Yes

5. Is the manuscript presented in an intelligible fashion and written in standard English?

Reviewer #1: Yes

Reviewer #2: Yes

6. Review Comments to the Author

Reviewer #1: The authors have adequately addressed all of both Revieweers' concerns in the text and additions to the tables.

Reviewer #2: I am largely satisfied with the authors’ changes and appreciate their efforts to respond to the reviewers’ comments so conscientiously. I note a few suggested clarifications for the authors.

On page 5, line 105, the authors refer to the method mix, but it appears that everything in lines 101-108 is prevalence of each method (the percentage of women using each method) rather than method mix (the percentage of use contributed by a particular method).

Optional: Also in this section, the authors not trends that are both substantial and likely statistically significant (e.g. increase of injectable use from 14.3% to 21.6% in 2003-2008/9) and those that are not (e.g. decrease of pill use from 7.5% to 7.2%). If the authors would like to streamline the paper, this is one paragraph where some modest edits could help achieve that.

On page 7, lines 149-150, the authors are technically correct that discontinuation while in need comprises women who discontinue while still at risk of becoming pregnant, do not want to become pregnant… However, isn’t this definition operationalized from the reasons for discontinuation (given in the second column of the calendar)? It would be helpful to clarify this and list which reasons comprise discontinuation while in need (e.g. husband disapproval, side effects, access, cost, wanted more effective method…) and those that do not comprise while in need (wanted to become pregnant, husband away/infrequent sex,…).

Thank you for clarifying that percentages in Table 4 do not sum to 100% because the table excludes discontinuation due to no further need. Could the table titled be revised to read, “Reasons for discontinuation while still in need” rather than “…among women in need,” as this change would be more accurate? And could the authors add a footnote to the tables similar to their response to the reviewer, e.g. “Percentages do not sum to 100% since women who discontinued while not in need of contraceptives, including those who reported they wanted to become pregnant/experienced menopause (2003, 42.1% and in 2014, 46.5%) were excluded from the analysis.”

7. PLOS authors have the option to publish the peer review history of their article (what does this mean?). If published, this will include your full peer review and any attached files.

Reviewer #1: No

Reviewer #2: **Yes: **Kerry MacQuarrie

---

## [Author Response · Author response to Decision Letter 1]

14 Oct 2020

Comments

1. On page 5, line 105, the authors refer to the method mix, but it appears that everything in lines 101-108 is prevalence of each method (the percentage of women using each method) rather than method mix (the percentage of use contributed by a particular method).

Response: This is noted. We have since updated the figures from line 98-106 and we are now reporting contraceptive method mix.

2. Optional: Also in this section, the authors not trends that are both substantial and likely statistically significant (e.g. increase of injectable use from 14.3% to 21.6% in 2003-2008/9) and those that are not (e.g. decrease of pill use from 7.5% to 7.2%). If the authors would like to streamline the paper, this is one paragraph where some modest edits could help achieve that.

Response: We take note of this comment. Ascertaining whether the changes observed in the trends of the method mix are statistically significant, is beyond the scope of our paper. We have however revised the write up to report on the changes observed without using the terms increase or decrease.

3. On page 7, lines 149-150, the authors are technically correct that discontinuation while in need comprises women who discontinue while still at risk of becoming pregnant, do not want to become pregnant… However, isn’t this definition operationalized from the reasons for discontinuation (given in the second column of the calendar)? It would be helpful to clarify this and list which reasons comprise discontinuation while in need (e.g. husband disapproval, side effects, access, cost, wanted more effective method…) and those that do not comprise while in need (wanted to become pregnant, husband away/infrequent sex,…).

Response: This is noted. We have since updated the write up to reflect the suggestion in lines 148-153

4. Thank you for clarifying that percentages in Table 4 do not sum to 100% because the table excludes discontinuation due to no further need. Could the table titled be revised to read, “Reasons for discontinuation while still in need” rather than “…among women in need,” as this change would be more accurate? And could the authors add a footnote to the tables similar to their response to the reviewer, e.g. “Percentages do not sum to 100% since women who discontinued while not in need of contraceptives, including those who reported they wanted to become pregnant/experienced menopause (2003, 42.1% and in 2014, 46.5%) were excluded from the analysis.”

Response: We have revised the title of the table as advised. We have also added a footnote.

---

## [Editor Report · Decision Letter 2]

19 Oct 2020

Patterns and determinants of modern contraceptive discontinuation among women of reproductive age: Analysis of Kenya Demographic Health Surveys, 2003–2014

PONE-D-20-19927R2

Dear Ms. Ontiri,

We’re pleased to inform you that your manuscript has been judged scientifically suitable for publication and will be formally accepted for publication once it meets all outstanding technical requirements.

Kind regards,

Catherine S. Todd

Academic Editor

PLOS ONE

Additional Editor Comments (optional):

I thank the authors for their attentive revisions and am pleased to accept this manuscript for publication.
---

## [Editor Report · Acceptance letter]

27 Oct 2020

PONE-D-20-19927R2 

Patterns and determinants of modern contraceptive discontinuation among women of reproductive age: Analysis of Kenya Demographic Health Surveys, 2003–2014 

Dear Dr. Ontiri:

I'm pleased to inform you that your manuscript has been deemed suitable for publication in PLOS ONE. Congratulations! Your manuscript is now with our production department. 

Kind regards, 

on behalf of

Dr. Catherine S. Todd 

Academic Editor

PLOS ONE